# Dietary Supplementation with a Cocoa–Carob Blend Modulates Gut Microbiota and Prevents Intestinal Oxidative Stress and Barrier Dysfunction in Zucker Diabetic Rats

**DOI:** 10.3390/antiox12081519

**Published:** 2023-07-29

**Authors:** Esther García-Díez, María Elvira López-Oliva, Francisco Perez-Vizcaino, Jara Pérez-Jiménez, Sonia Ramos, María Ángeles Martín

**Affiliations:** 1Instituto de Ciencia y Tecnología de Alimentos y Nutrición (ICTAN-CSIC), 28040 Madrid, Spain; esther.garciad@ictan.csic.es (E.G.-D.); jara.perez@ictan.csic.es (J.P.-J.); s.ramos@ictan.csic.es (S.R.); 2Departamento de Fisiología, Facultad de Farmacia, Universidad Complutense de Madrid, 28040 Madrid, Spain; elopez@farm.ucm.es; 3AFUSAN Group, Sanitary Research Institute of the San Carlos Clinical Hospital (IdISSC), 28040 Madrid, Spain; 4Departamento de Farmacología y Toxicología, Facultad de Medicina, Universidad Complutense de Madrid, 28040 Madrid, Spain; fperez@med.ucm.es; 5CIBER de Enfermedades Respiratorias (CIBERES), Instituto de Salud Carlos III (ISCIII), 28029 Madrid, Spain; 6Instituto de Investigación Sanitaria Gregorio Marañón (IISGM), 28007 Madrid, Spain; 7CIBER de Diabetes y Enfermedades Metabólicas Asociadas (CIBERDEM), Instituto de Salud Carlos III (ISCIII), 28029 Madrid, Spain

**Keywords:** type 2 diabetes, cocoa, carob, phenolic metabolites, gut microbiota, gut barrier, redox status, inflammation

## Abstract

We have recently developed a cocoa–carob blend (CCB) rich in polyphenols with antidiabetic properties. In this study, we investigated whether its benefits could be related to gut health and gut microbiota (GM) composition and the likely phenolic metabolites involved. Zucker diabetic fatty rats were fed on a standard or a CCB-rich diet for 12 weeks. Intestinal barrier structure and oxidative and inflammatory biomarkers were analyzed in colonic samples. GM composition and phenolic metabolites were evaluated from feces. The results show that CCB improved mucin and tight-junction proteins and counteracted gut oxidative stress and inflammation by regulating sirtuin-1 and nuclear factor erythroid 2-related factor 2 (Nrf2) levels. CCB also modulated the composition of the GM, showing increases in *Akkermansia* and *Bacteroides* and decreases in *Ruminococcus* genera. Correlation analysis strengthened the associations between these genera and improved pathological variables in diabetic animals. Moreover, 12 phenolic metabolites were identified in CCB feces, being2,3-dihydroxybenzoic and 3,4,5-trihydroxybenzoic acids significantly associated with increased levels of *Akkermansia* and *Oscillospira* genera. Our findings support the potential use of CCB to prevent intestinal damage and dysbiosis in T2D, which would help to delay the progression of this pathology.

## 1. Introduction

Type 2 diabetes (T2D) is a complex metabolic disorder considered a public health issue worldwide due to its high prevalence and associated complications [1]. In recent years, the role of the intestine in the development and progression of T2D has been increasingly recognized [2]. Oxidative stress induced by the diabetic milieu leads to decreased expression of epithelial tight-junction (TJ) proteins in the intestine, altering its barrier function and increasing local and systemic inflammation, which promotes the progression of this metabolic disease [3]. In addition, the gut microbiota (GM), composed of many microbial species that modulate gut barrier integrity and metabolism, is disrupted in T2D, which contributes to alterations in the gut’s permeability, inflammation, and glucose metabolic dysfunctions [4,5]. Therefore, strategies that target oxidative stress and the GM might be promising in attenuating intestinal barrier damage and could be a potential way to help control T2D.

Polyphenols are a group of vegetable-derived compounds with health-promoting properties [6], for which the GM plays a key role through a bidirectional interaction [7]. On the one hand, polyphenols modulate the GM composition through prebiotic-like mechanisms, favoring the growth of beneficial bacteria and increasing GM diversity [8]. On the other hand, the GM is responsible for the biochemical transformation of polyphenols into a large array of smaller phenolic compounds, usually more active and better absorbed than the parent compounds [9]. Accordingly, PPs are emerging as an attractive option in interventions focused on the recovery of a healthy intestinal ecology.

Cocoa constitutes one of the richest sources of dietary polyphenols, with proven effects against diabetes and its associated complications [10]. Cocoa-rich diets have exerted their ability to modulate the GM in healthy rats [11,12] and humans [13], primarily due to the high polyphenol content. More importantly, cocoa can modify GM composition in diabetic rats, and these changes have been closely associated with improved glucose homeostasis [14]. Likewise, carob, a Mediterranean legume with a high content of polyphenols, has also demonstrated beneficial properties against T2D [15,16]. 

In line with this, we recently developed a potential functional food combining cocoa powder and carob flour to obtain a cocoa–carob blend (CCB) [17] that has demonstrated antidiabetic and cardioprotective effects [18]. This functional mixture is especially rich in polyphenols (particularly nonextractable proanthocyanidins), which makes it a potential prebiotic for ameliorating intestinal dysbiosis and producing bioactive microbial-derived phenolic metabolites. With this in mind, in the present work, we aimed to investigate whether the antidiabetic effects of CCB could be related to its potential effects on gut health and GM composition, as well as the likely phenolic metabolites involved in these processes. 

## 2. Materials and Methods

### 2.1. Materials and Chemicals

Nicotine adenine dinucleotide reduced salt, nicotine adenine dinucleotide phosphate reduced salt (NADPH), 2′,7′-dichlorofluorescin diacetate (DCFH), reduced glutathione (GSH), glutathione reductase (GR), *tert*-butylhydroperoxide, o-phthaldehyde, streptavidin–biotin conjugated horseradish peroxidase (HRP), 3,3′-diaminobenzidine (DAB), and glucose were purchased from Sigma Chemical (Madrid, Spain). Antiproliferative cell nuclear antigen (PCNA; PC-10) was purchased from Lab Vision Corporation and Bionova- Científica SL. Anti-SIRT1 (sc-74465), anti-occludin (sc-5562), anti-ZO-1 (sc-33725), anti-TNF-α (sc-52746), and anti-IL6 (sc-57315) were purchased from Santa Cruz Biotechnology (Quimigen, Madrid, Spain). Antiphospho-Nrf2 (#12811) was purchased from Signalway antibody (Quimigen, Madrid, Spain). Terminal transferase recombinant, biotin-16-dUTP, and proteinase K were purchased from Roche Applied Science (Roche Diagnostic, GMbH, Madrid, Spain).

### 2.2. Cocoa–Carob Blend Diet

CCB was a mix of pure cocoa powder (a kind gift from Idilia S.L., Barcelona, Spain) and carob flour (Casa Ruiz Granel Selecto S.L., Madrid, Spain) in a proportion of 60:40. The final product contained 16.7% polyphenols. Further information about the composition of CCB is provided elsewhere [17]. The CCB-rich diet (10%) was elaborated by adding 100 g/kg of CCB to AIN-93G diet. Control and CCB diets were isoenergetic. In our experiment, animals received a dose of 237 mg/kg/day of extractable polyphenols. Using the body surface area (BSA) normalization method [19], the amount of CCB employed in this animal study is approximately equivalent to a dose of 2305 mg of extractable polyphenols/day for a 60 kg adult human. 

### 2.3. Animals and Experimental Design

The Ethics Committee from Comunidad de Madrid (PROEX 079/19) approved the protocol for animal experimentation. Animals were treated according to the European (2010/63/EU) and Spanish (RD 53/2013) legislation on care and use of experimental animals.

Male Zucker diabetic fatty rats (ZDF) and their Zucker lean (ZL) controls were purchased from Charles River Laboratories (L’arbresle, France) at 11 weeks of age. All animals were acclimated under standard controlled conditions (21 ± 1 °C; 50–60% humidity and 12 h day/night cycle). One week later, ZDF rats were randomly sorted into two groups; the first one received a standard AIN-93G diet (ZDF) and the second one received a cocoa–carob blend (CCB)-rich diet (10%) (ZDF-CCB). The ZL nondiabetic group received the standard AIN-93G diet (ZL). Animals were given ad libitum access to food and water. At 24 weeks of age (12 weeks of supplementation), animals were sacrificed, and blood samples were collected for biochemical analysis. The entire colon was rapidly resected for histological and biochemical analyses. One day before the sacrifice, fresh fecal samples were collected using abdominal massage in sterilized tubes and immediately frozen in liquid nitrogen and stored at −80 °C for further analyses.

### 2.4. Biochemical Determinations

Blood glucose was determined using an accounted glucose analyzer (LifeScan España, Madrid, Spain). Serum insulin and Hb1Ac were analyzed with an ELISA kit (Rat Insulin, Mercodia, Uppsala, Sweden; HbA1c Kit Spinreact, BioAnalitica, Madrid, Spain). To assess insulin resistance and secretion, fasting glycemia and insulinemia were used to calculate the homeostatic model assessment of insulin resistance (HOMA-IR) and secretion (HOMA-B) using the following formulas, HOMA-IR = fasting insulin (mU/mL) × fasting glucose (mM)/22.5 and HOMA-B = 20 × fasting insulin (mU/mL)/[fasting glucose (mM) − 3.5], respectively.

One week before the animals were sacrificed, a glucose tolerance test (GTT) was conducted. Following an overnight fast, rats were administered 35% glucose solution intraperitoneally (2 g/kg of body weight). Blood samples were collected from the tail vein before the glucose load (t = 0) and at 30, 60, 90, and 120 min after glucose administration. Glucose levels were determined using a glucometer (LifeScan España, Madrid, Spain). Changes in glucose were calculated as the area under the curve (AUC) above the basal levels.

### 2.5. Histological and Immunohistochemical Analyses of Colon

Sections from the most distal portion of the colon were processed and embedded in paraffin for histological and immunohistochemical analyses. Sections were stained with hematoxylin–eosin (H&E), periodic acid–Schiff (PAS) and Masson’s trichrome staining. Images of the stained sections were captured under light microscopy using a digital Leica DFC 320 camera (Leica, Madrid, Spain) and quantified with ImageJ software (Fiji image J; 1.52i, NIH, Bethesda, MD, USA). Crypt density and depth were measured as crypt number per millimeter and cell number per hemicrypt of the distal colon mucosa, respectively. The expression level of the mucin glycoprotein was calculated as the number of PAS-positive cells per crypt. Collagen deposition was assessed by quantifying the percentage of positive Masson’s trichrome-stained area in total area.

For the immunohistochemical staining and histochemical staining, the colonic sections were deparaffinized, and endogenous peroxidase activity was quenched. Next, the sections were incubated with the primary antibodies overnight at 4 °C. Secondary antibodies were used to detect primary antibodies, followed by streptavidin-tagged horseradish peroxidase, and visualized using 3,3′- diaminobenzidine (DAB) substrate (Sigma Chemical, Madrid, Spain). The sections were counterstained with Harris’s hematoxylin, dehydrated and mounted. In the stained sections, the specific protein immunostaining appeared as a brown color, while the nuclear staining with hematoxylin was light blue. At least 20 perpendicular well-oriented crypts were examined in each animal under light microscopy. The proliferative labeling index (LI) (%) was calculated as the number of positive nuclei × 100/total number of cells per crypt column height. PCNA, ZO-1, occludin, SIRT-1, pNrf2, TNF-α, and IL-6 protein levels were evaluated as percentage of the stained area to the total area per crypt by using ImageJ v1.52j software.

### 2.6. Terminal Deoxynucleotidyl Transferase dUTP Nick End Labeling (TUNEL) Assay

Apoptotic colonic cells were detected in situ by using the terminal deoxynucleotidyl transferase UTP nick end labeling (TUNEL) assay [14]. The apoptotic index was determined by calculating the proportion of cells undergoing apoptosis within a crypt column at a magnification of 400×. Then, the number of TUNEL-positive cells was divided by the total number of cells counted within 50 well-oriented crypts that spanned the full length of the section.

### 2.7. Determination of ROS

ROS were quantified in colon homogenates using the DCFH assay based on the oxidation of dichlorofluorescein (DCF) [20]. Fluorescence was measured in a microplate reader at an excitation wavelength of 485 nm and an emission wavelength of 530 nm (Bio-Tek, Winooski, VT, USA).

### 2.8. Protein Carbonyl Levels

Protein oxidation of colon homogenates was measured as carbonyl levels [20]. Colon samples were homogenized and derived with 2,4 dinitrophenylhydrazine (DNPH). Absorbance was measured at 360 nm and carbonyl content was expressed as nmol/mg protein using an extinction coefficient of 22,000 nmol L^−1^ cm^−1^.

### 2.9. GSH Levels and GPx and GR Activities

The content of reduced glutathione (GSH) was quantitated using a fluorometric assay [15] based on the reaction of GSH with o-phthalaldehyde (OPT) at pH 8.0, which generates fluorescence. The fluorescence was measured at 460/340 mm excitation and emission wavelengths, respectively. The activity of GPx was assessed via the oxidation of GSH by GPx, using *tert*-butylhydroperoxide as substrate, coupled to the disappearance of nicotinamide adenine dinucleotide phosphate (NADPH) caused by GR [20]. GR activity was determined by following the decrease in absorbance due to the oxidation of NADPH utilized in the reduction of oxidized GSH [20].

### 2.10. DNA Extraction and 16S Gene PCR Amplification

G-spin columns (INTRON Biotechnology, Seongnam, Republic of Korea) were used for the DNA extraction from fecal samples. DNA concentration was determined using Quant-IT PicoGreen reagent (ThermoFisher Scientific, Inc., Waltham, MA, USA). DNA extracted (3 ng) was used to amplify the V3–V4 region of 16S rRNA gene [21]. The quality and integrity of individual amplicon libraries were subjected to analysis using a Bioanalyzer 2100 (Agilent, Santa Clara, CA, USA). DNA samples were sequenced at the Unidad de Genómica (Parque Científico de Madrid, Madrid, Spain) on an Illumina MiSeq instrument (Illumina, San Diego, CA, USA). Operational taxonomic units (OTUs) were assigned using the 16S-metagenomics workflow (version 1.0.1) associated with the Base Space Hub provided by Illumina (Illumina, 2013 version). For OTU classification, an Illumina-curated version of the GreenGenes taxonomic database was utilized, which implements the Ribosomal Database Project (RDP) Classifier [22]. The Taxonomy Database (National Center for Biotechnology Information, Bethesda, MD, USA) was used for classification and nomenclature. We normalized the reads in each sample for each genus and phylum to total genus and phylum reads of that sample. Only taxa with a percentage of reads >0.2% were used for this analysis.

### 2.11. Determination of Phenolic Metabolites in Feces

An aqueous extraction, which was adapted from a reported protocol [23], was performed on the freeze-dried fecal samples. Five hundred mg of fresh feces were homogenized with 5 mL of a saline solution (0.9% NaCl) and centrifuged at 5000 rpm for 30 min. The supernatant was passed through 0.45 µm filters and used for analysis.

Targeted metabolomics was applied for identifying the metabolites in feces resulting from CCB-rich diet intake by using liquid chromatography coupled to a mass spectrometer with electrospray ionization and a quadrupole/time-of-flight mass analyzer (HPLC-ESI-QTOF MS) (Agilent 1200, Agilent Technologies, Santa Clara, CA, USA). The column used was a 250 mm × 4.6 mm i.d., 5 μm, Gemini C18 (Phenomenex, Torrance, CA, USA). Gradient elution was carried out with a binary system consisting of 0.1% formic acid in acetonitrile (A) and 0.1% aqueous formic acid (B). Data were acquired using negative (for phenolic compound metabolites) and positive (for methylxanthine metabolites) ion modes with a mass range of 100−1200 Da, using a source temperature of 325 °C and a gas flow of 10 L/h. For identification, the molecular formula proposed by MassHunter Workstation 162 software version 4.0 for the different signals was compared with previously reported phenolic compounds in a cocoa–carob blend [17] as well as with existing bibliography on polyphenol metabolites, and a maximum error of 10 ppm was accepted.

For quantification, a standard calibration curve was built for each compound, although in some cases semiquantification was performed using 3-hydroxyphenylpropionic acid for dihydroxyphenylvalerolactones; phenylvaleric acid for phenylvaleric acids and hydroxyphenylpropionic acids; 3-hydroxyphenylacetic acid for hydroxyphenylacetic acids and derivatives; 3,4-dihydrozybenzoic acid for hydroxybenzoic acids; and gallic acid for its derived metabolites.

### 2.12. Statistical Analysis

Statistical analysis was performed using the GraphPad Prism software (Prism version 8 for Windows, GraphPad Software, La Jolla, CA, USA). The Shapiro–Wilk normality test was used to check if the data followed a Gaussian distribution. For normally distributed data, one-way analysis of variance (ANOVA) was performed. Statistically significant differences between means were determined using the Tukey post hoc test. A two-tailed *p* < 0.05 was considered significant. All data are expressed as mean ± standard deviation.

For targeted metagenomics, since normality was not accomplished, the nonparametric Kruskal–Wallis and Mann–Whitney U tests were applied to analyze significant differences between groups. The Kruskal–Wallis test followed by the Mann–Whitney U test were used to compare all the different pairs of treatments. A two-tailed *p* < 0.05 was considered significant. Results are expressed as mean concentrations (nmol/g fresh feces) with standard deviation.

For metagenomic analysis, the Shannon, Chao1, and Pielou evenness indices were calculated to analyze α-diversity using Past software (ver3.21, Oslo, Norway). Principal components analysis (PCA) was also carried out with Past software (ver4.04, Oslo, Norway). Normal distribution of the variables was confirmed, and genera and phyla were compared using a two-way ANOVA. Multiple comparisons were carried out using two-stage linear step-up procedure of Benjamini, Krieger, and Yekutieli to correct for false discovery rate. Differences were considered significant when corrected *p* values, i.e., q values, were <0.05.

The relationships between biochemical parameters, GM, and phenolic metabolites were evaluated using Spearman correlations. *p*-values of 0.05 were considered statistically significant. A heatmap was used to show the correlation analysis.

## 3. Results and Discussion

In this study, we tested the effects of a 12-week supplementation with a cocoa–carob blend (CCB) on gut health in an animal model of T2D, Zucker diabetic fatty (ZDF) rats. In addition, the compositional changes in the GM induced by CCB intake and the production of key phenolic metabolites were also investigated.

As shown in Table 1, the daily diet intake was found to be significantly higher in the ZDF groups compared with the nondiabetic ZL animals, confirming their hyperphagic condition. However, at the end of the study, the body weight gain was decreased in the ZDF groups, which is indicative of their diabetic status. In terms of glucose homeostasis, the ZDF rats exhibited a significant increase in fasting and postprandial glucose levels, as well as elevated glycosylated hemoglobin (HbA1c) values. Moreover, the ZDF rats demonstrated glucose intolerance, as evidenced by an increased area under the curve (AUC) during the GTT. Interestingly, the supplementation of the diet with CCB resulted in significant improvements in all these parameters.

### 3.1. CCB Supplementation Preserves the Structure and Integrity of the Intestinal Barrier in the Colon Mucosa of Diabetic Animals

Evidence suggests that impaired gut barrier integrity is an important pathogenic process contributing to the progression of T2D [2]. Indeed, both animal models and T2D patients display modifications in the gut mucosal structure and higher intestinal permeability [24,25]. Here we observed that chronic intake of CCB was able to ameliorate the altered colonic mucosal morphology in diabetic animals. In particular, the ZDF-CCB rats showed deeper crypts, with significantly higher numbers of goblet cells and less fibrosis than the ZDF animals (Figure 1a–c). Mucin proteins secreted by goblet cells are vital components of the biochemical barrier against gut microbes [26] and contribute to strengthening the integrity of the intestinal barrier. Herein, colonocyte proliferation and apoptosis were significantly increased in ZDF-CCB rats, indicating that CCB intake induced a faster renewal rate of the colonic epithelium (Figure 1d). At the same time, TJ proteins such as zonula occludens 1 (ZO-1) and occludin constitute the physical barrier regulating gut permeability and the entry of intestinal content [25]. In the present study, levels of TJ proteins were significantly reduced in the colonic epithelia of diabetic animals, whilst CCB supplementation remarkably increased them (Figure 1e). Altogether, our results indicate that a regular intake of CCB preserves the structure and integrity of the intestinal barrier in the colon mucosa of diabetic animals. Moreover, these results are consistent with current studies evidencing that dietary polyphenols may enhance intestinal barrier function by increasing the levels of TJ proteins [27].

### 3.2. CCB Protects the Guts of Diabetic Animals from Oxidative Damage and Inflammation

A diabetic milieu increases gut oxidative stress and inflammation, playing a key role in the pathogenesis of intestinal disorders, including intestinal barrier disruption [3]. In this sense, long-term intake of CCB was effective in reducing gut oxidative damage by decreasing ROS levels and oxidative injury, and increasing GSH and its related antioxidant enzymes, GPx and GR, in the colons of diabetic animals (Figure 2a). Likewise, CCB also decreased the levels of the proinflammatory cytokines TNF-α and IL-6 (Figure 2b), indicating that modulation of oxidative stress and inflammation could be one of the mechanisms for CCB to maintain the barrier integrity and gut health in diabetes. Next, we evaluated the phosphorylated levels of the nuclear transcription factor erythroid 2-related factor 2 (Nrf2), which modulates both antioxidant enzyme expression and the inhibition of inflammatory response [28]. Furthermore, Nrf2 activation has recently been associated with increased expression of epithelial TJ proteins and maintenance of intestinal barrier integrity [29,30]. Accordingly, we found that CCB significantly prevented the decreases in levels of pNrf2 induced by diabetes (Figure 2c), further supporting our results about beneficial effects against oxidative stress and inflammation and TJ protein levels. Likewise, CCB supplementation also prevented the depletion of sirtuin-1 (SIRT1) (Figure 2d), a histone deacetylase protein that has been identified as a negative regulator for oxidative stress by targeting Nrf2 [31]. Multiple pieces of evidence have indicated that polyphenols and polyphenol-rich foods are potent activators of SIRT1. In particular, we have previously shown that both cocoa [20] and CCB [18] increase the reduced levels of SIRT1 in the arteries and hearts of diabetic animals, respectively. More importantly, a very recent study has demonstrated that polyphenol pterostilbene prevents endoplasmic reticulum stress and intestinal barrier damage via SIRT-dependent mechanisms [32]. Collectively, these findings seem to indicate that CCB could protect the guts of diabetic animals from oxidative damage and inflammation via Nrf2/SIRT1 signaling. Given the important role of gut homeostasis in glucose regulation in T2D, we suggest that ameliorating the intestinal mucosal injury might be a potential way for CCB to decrease the progression of diabetes in ZDF rats [18].

### 3.3. CCB Modifies the Structure and the Composition of Gut Microbiota in Diabetic Animals

There is extensive evidence indicating that GM dysbiosis is closely related to alterations in both the structure and the function of the intestine, promoting the development of metabolic events [33]. Thus, we hypothesized that the potential prebiotic activity of CCB could be one of the mechanisms behind its protective effect in the colons of diabetic animals. In order to investigate the effect of CCB on GM composition in diabetic animals, we next performed a metagenomic analysis of fecal samples. As expected, diabetes affected the structure of the GM, resulting in decreased species richness and diversity (indicated by the Shannon, Simpson, and Chao indices) (Figure 3a). Furthermore, the composition of the microbial communities was different among the three experimental groups (Figure 3b). Diabetic animals showed an increase in the relative abundance of *Firmicutes phylum* along with a decrease in the abundance of *Bacteroidetes*, even though no significant differences were found in the *Firmicutes/Bacteroidetes* (F/B) ratio (Figure 3c,d). However, CCB supplementation significantly restored GM composition in diabetic animals, which could be linked to the positive improvements in gut health and glucose homeostasis found in this group.

Specifically, at the genus level, CCB supplementation largely prevented the increase in the relative abundance of *Lactobacilus* and *Phascolarctobacterium* and the decrease in *Oscillospira genera* found in diabetic animals (Figure 4). More importantly, CCB intake augmented the abundance of beneficial genera such as *Akkermansia* and *Bacteroides* and decreased that of the opportunistic genus *Ruminococcus* in diabetic animals. Numerous studies have stablished a solid association between the consumption of polyphenol-rich foods and subsequent enrichment of *Akkermansia* and *Bacteroides genera* [34]. Actually, the *Akkermansia* genus plays an important role in promoting gut barrier integrity, modulating the immune response, and inhibiting gut inflammation [35]. Likewise, *Bacteroides* has been found to upregulate the expression of TJ genes, reduce gut permeability and improve glucose metabolism [36].

Spearman’s correlation analyses performed between these genera and biochemical biomarkers related to diabetes support that GM modifications induced by CCB are associated with positive changes in intestinal barrier integrity and gut inflammation (Figure 5). In addition, the *Bacteroides* and *Phascolarctobacterium* genera were significantly correlated with improved fasting and postprandial glucose values. However, no significant association was observed with HbA1c (a long-term measure of glucose control). It is worth noting that previous studies have reported inconsistent or conflicting findings regarding the association between specific bacterial groups and HbA1c levels [37,38,39]. This suggests that the relationship between gut microbiota and glycemic control is complex and likely influenced by various factors, such as individual variations [37], the development stages of diabetes [38], and the inherent heterogeneity of the gut microbiota itself [39]. Altogether, we suggest that the beneficial effects of CCB on gut homeostasis could be partly due to the increase in the potentially health-promoting bacterial genera *Akermansia* and *Bacteroides*. These results are also in agreement with the polyphenol duplibiotic effect proposed by Rodriguez-Daza et al. [40], showing the ability of PPs to modulate the GM by both prebiotic and antimicrobial modes of action.

### 3.4. Association between Polyphenol-Derived Metabolites from CCB and Gut Microbiota Modification

Importantly, polyphenols can also be metabolized by the GM into small phenolic bioactive metabolites that are important players in maintaining gut health and glucose metabolism [41]. Therefore, our next objective was to search for potential interactions between GM genera and polyphenol-derived metabolites. A targeted metabolomics approach was performed, based on the polyphenolic profile in CCB, namely consisting of proanthocyanidins, hydrolyzable tannins, phenolic acids, monomeric flavonoids (described in [17]), and the reported main metabolites for these phenolic classes. A total of 12 metabolites were identified (Table 2). They correspond to the families of valerolactones, phenylvaleric acids, phenylpropionic acids, phenylacetic acids, and benzoic acids. These families come from the successive degradation of proantochyanidins, although some of these metabolites are common and are found in the degradation pathways of other polyphenols found in CCB, such as quercetin [42]. Moreover, 3,4,5-hydroxybenzoic acid (gallic acid, GA) may also be derived from hydrolyzable tannins, as reported in carob [15]. Some of these metabolites appeared in their native forms, while others were detected as glucuronidated or sulfated derivatives, indicating a hepatic transformation after absorption, followed by enterohepatic circulation. Spearman’s correlation analysis between the identified phenol-derived metabolites and the significant genera modulated by CCB showed a positive association of the *Oscillospira* genus with the phenolic metabolites 2,3-dihydroxybenzoic acid (DHBA) and GA (Figure 6). Moreover, increased levels of *Akkermansia* were also positively correlated with GA. Notably, DHBA has revealed potential antidiabetic effects by reducing oxidative stress and inflammation in both endothelial [43] and renal cells [44]. Furthermore, GA has been shown to preserve the expression levels of TJ proteins in LPS-stimulated intestinal IPEC-J2 cells [45] and to reduce both oxidative stress and inflammation in the colons of animals with DSS-induced colitis by activating the Nrf2 pathway [46]. More recently, it has also been demonstrated that GA has the ability to induce the SIRT1/Nrf2 signaling pathways in HepG2 cells [47]. In agreement with this, here we propose that DHBA and GA may act locally in the colons of diabetic animals by modulating the SIRT1/Nrf2 pathway to reinforce the intestinal barrier structure and function. Taken together, our results seem to indicate that CCB supplementation may protect the gut barriers of diabetic animals by ameliorating intestinal inflammation and oxidative stress through its effects on GM composition and its phenolic metabolites. In addition, it is interesting to note that these phenolic metabolites could also modulate the GM by themselves [48] and be absorbed to exert their beneficial effects on other target tissues related to glucose metabolism [9,42].

Due to the limited identification of potential genera involved in the production of phenol-derived metabolites in in vivo T2D models, the results of this study are novel and significant. They provide a theoretical basis for understanding the relationship between specific genera and phenol-derived metabolites, which may contribute to the beneficial effects of CCB polyphenols on gut health in the context of T2D. Nonetheless, this study has some limitations. Quantification of phenolic metabolites in plasma would be valuable to gain better insights into the biological health effects of phenol-derived metabolites. Additionally, while correlation studies are important for identifying close associations between GM modifications and polyphenolic health effects, they do not establish cause–effect relationships. In this regard, future studies utilizing fecal transplantation from CCB-fed donors could provide valuable information. Furthermore, the application of metabolomics to the microbiota could yield a more comprehensive analysis, potentially advancing our understanding of the ultimate causality underlying these effects.

## 4. Conclusions

In conclusion, the present work demonstrates for the first time that regular consumption of a cocoa–carob blend rich in polyphenols prevents intestinal oxidative stress and inflammation and improves barrier integrity in the colons of diabetic animals. This protective effect occurs in association with the ability of CCB to modulate GM composition, promoting the growth of beneficial bacteria genera (*Akkermansia* and *Bacteroides*) and inhibiting the pathogenic ones (*Ruminococcus*). Moreover, correlation analysis reinforced the associations between these genera and improved pathological parameters in diabetic animals and provided valuable information on potential keystone bacterial genera involved in the production of bioactive phenolic metabolites (GA and DHBA). Our study further suggests that key phenolic metabolites associated with CCB intake play an essential role in this protective effect, and the modulation of the Nrf2/SIRT1 pathway is proposed as a likely mechanism.

Therefore, polyphenol-rich foods such as CCB might represent an interesting option to prevent T2D progression through the potential of PPs to exert a prebiotic impact on the GM. Future human clinical trials with T2D patients are needed to confirm this preventive effect.

## Figures and Tables

**Figure 1 antioxidants-12-01519-f001:**
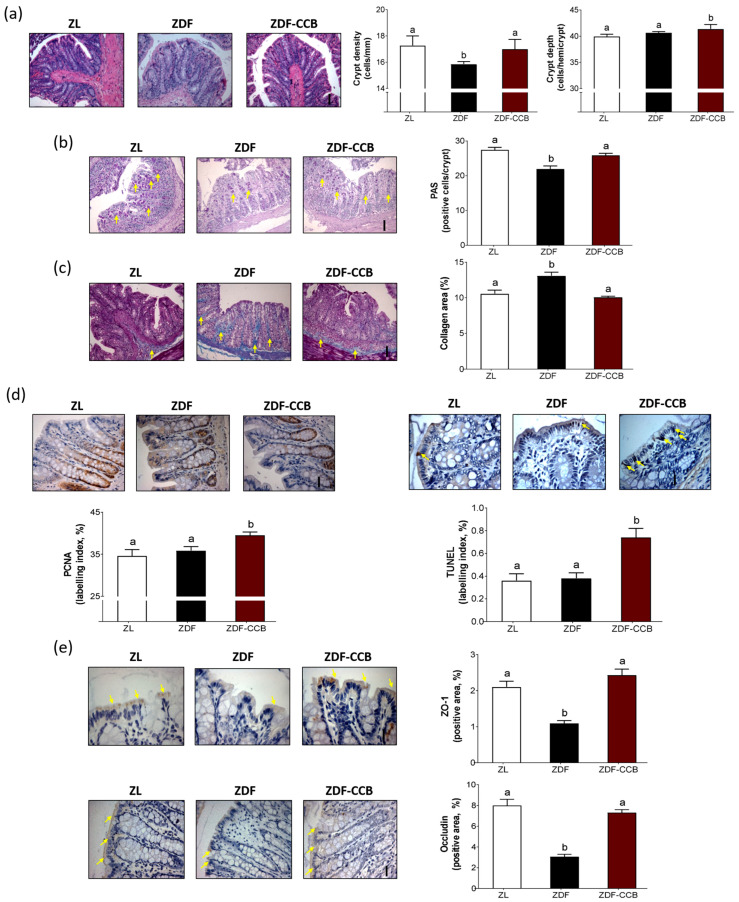
Distal colonic mucosa structure. (**a**) Representative hematoxylin–eosin (H&E)-stained sections and crypt depth (cell number per hemicrypt) and crypt density (number of crypts/mm) (scale bar: 50 µm). (**b**) Representative images of mucin glycoprotein revealed with PAS staining (arrows indicate magenta-positive cells) and quantitative analysis of positive PAS staining cells (%) (scale bar: 50 µm). (**c**) Representative photographs of collagen with Masson’s trichrome stain (arrows indicate blue-positive areas) (scale bar: 50 µm) and quantitative analysis of collagen-positive area (%). (**d**) Representative photographs of immunohistochemical staining of proliferating cell nuclear antigen (PCNA) (brown-positive nuclei) and PCNA labeling index (%) (scale bar: 20 µm). Colonic epithelial apoptosis as revealed using TUNEL assay (arrows indicate brown-positive nuclei) and TUNEL labeling index (%) (scale bar: 2.5 µm). (**e**) Representative immunohistochemical photographs with arrows indicating ZO-1 and occludin brown-marked cells in endothelial junctional regions and positive area staining (%) (scale bars: 2.5 and 20 µm, respectively). Means ± SD of 6–8 samples per condition. Means without a common letter differ significantly (*p* < 0.05). Zucker lean rats (ZL), Zucker diabetic rats (ZDF), Zucker diabetic rats fed with a CCB-rich diet (ZDF-CCB).

**Figure 2 antioxidants-12-01519-f002:**
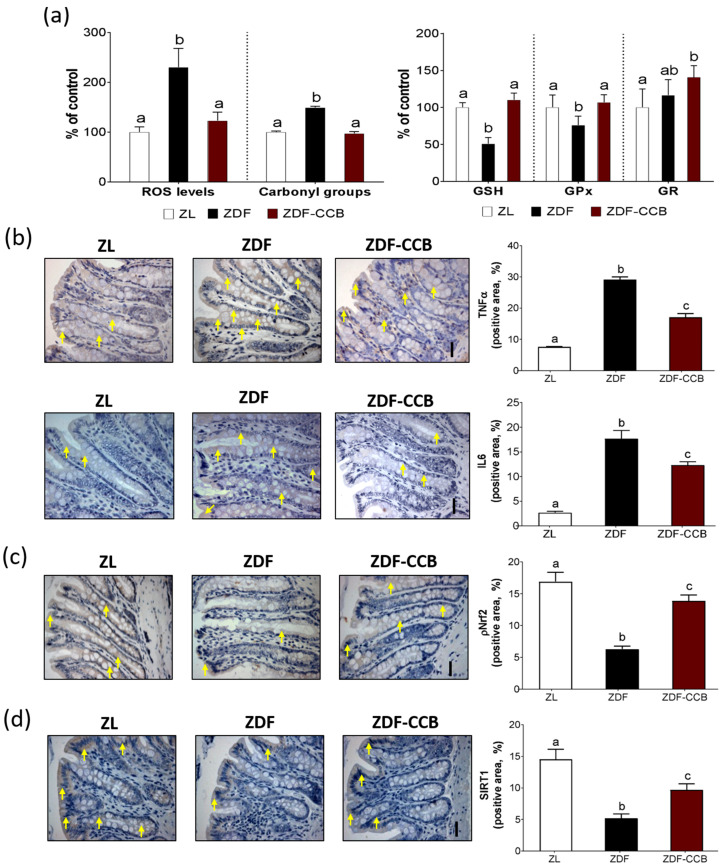
Markers of oxidative stress and inflammation in colon. (**a**) Percentage levels of ROS and carbonyl groups relative to the control condition (ZL). Percentage levels of glutathione (GSH) and antioxidant enzymes glutathione peroxidase (GPx) and glutathione reductase (GR) relative to the control ZL condition. (**b**) Representative immunohistochemistry images of TNF-α and IL-6 in distal colonic mucosa (arrows indicate brown-positive cells) and positive area staining (%) (scale bar: 20 µm). (**c**) Representative immunohistochemical images of pNrf2 protein expression in distal colonic mucosa (arrows indicate brown-positive cells) and positive area staining (%) (scale bar: 20 µm). (**d**) Representative immunohistochemical images of SIRT1 protein expression in distal colonic mucosa (arrows indicate brown-positive cells) and positive area staining (%) (scale bar: 20 µm). Means ± SD of 6–8 samples per condition. Means without a common letter differ significantly (*p* < 0.05). Zucker lean rats (ZL), Zucker diabetic rats (ZDF), Zucker diabetic rats fed with a CCB-rich diet (ZDF-CCB).

**Figure 3 antioxidants-12-01519-f003:**
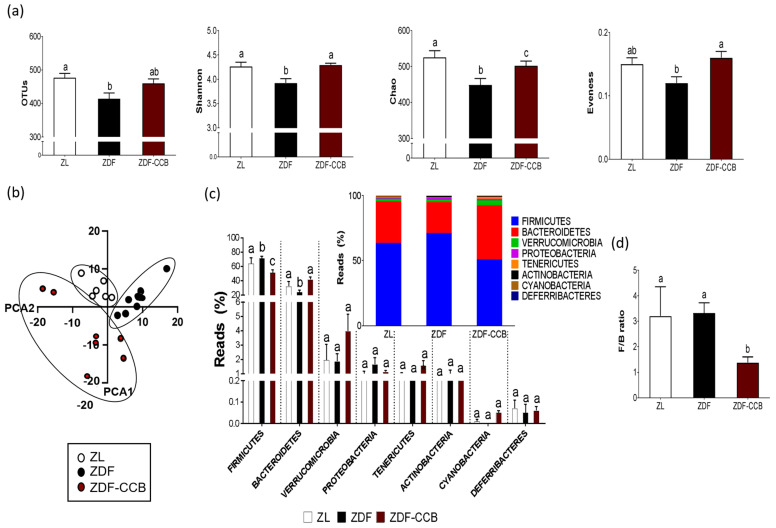
Bacterial diversity and taxa composition of microbial population. (**a**) Alpha diversity indices include total operational taxonomic units (OTUs), Shannon, Chao1, and evenness indices. (**b**) Beta diversity analyzed with unsupervised principal component analysis with all OTUs detected. (**c**) Relative abundance of the most abundant phylum; the inset shows the stacked column chart. (**d**) *Firmicutes/Bacteroidetes* ratio (F/B ratio). Data represent means ± SD of 6–8 animals per condition. Means sharing the same letter are not significantly different from each other (*p* < 0.05). Zucker lean rats (ZL), Zucker diabetic rats (ZDF), Zucker diabetic rats fed with a CCB-rich diet (ZDF-CCB).

**Figure 4 antioxidants-12-01519-f004:**
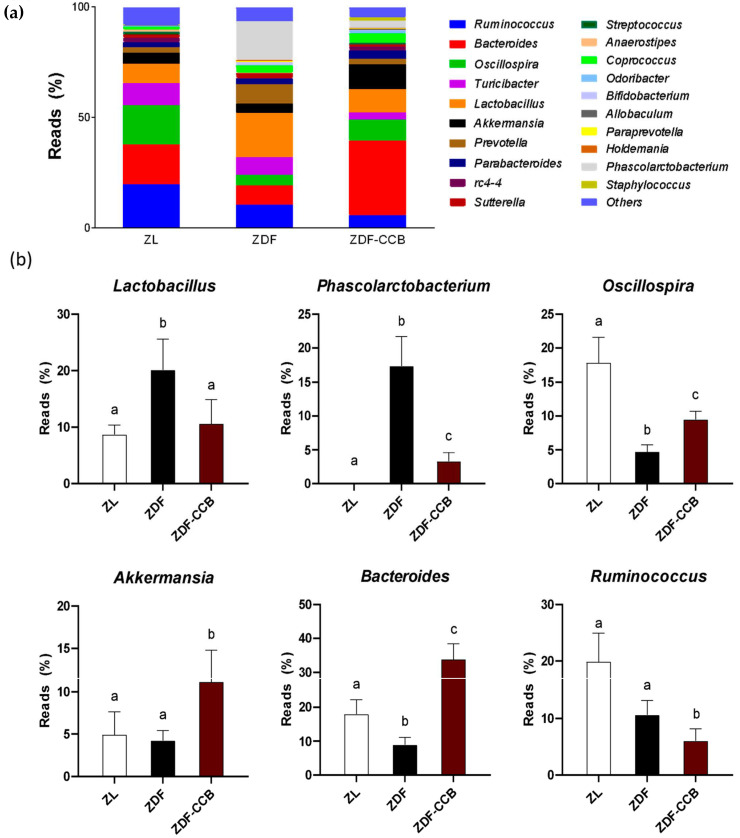
Relative abundance of microbial population at the genus level. (**a**) Distribution bar plot of genera with relative abundance greater than 0.2%. (**b**) Composition of the most abundant bacterial genera modified with diabetes or with CCB expressed as a percent of total bacteria. (Data represent means ± SD of 6–8 animals per condition. Means sharing the same letter are not significantly different from each other (*p* < 0.05). Zucker lean rats (ZL), Zucker diabetic rats (ZDF), Zucker diabetic rats fed with a CCB-rich diet (ZDF-CCB).

**Figure 5 antioxidants-12-01519-f005:**
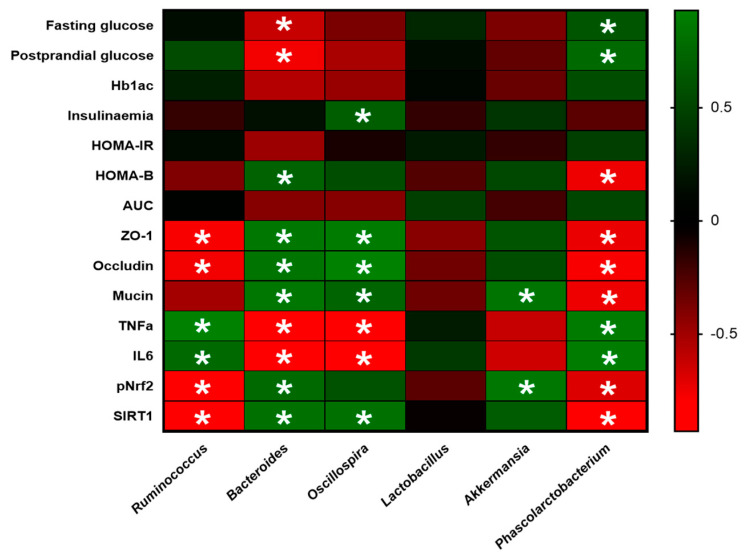
Interactions between gut microbiota genera and physiological parameters. Heatmap of correlation between the main significantly altered genera in the gut microbiota and host parameters related to diabetes and intestinal integrity and inflammation. Spearman correlation values were used for the matrix. The intensity of the color represents the degree of association. * Denotes adjusted *p* < 0.05.

**Figure 6 antioxidants-12-01519-f006:**
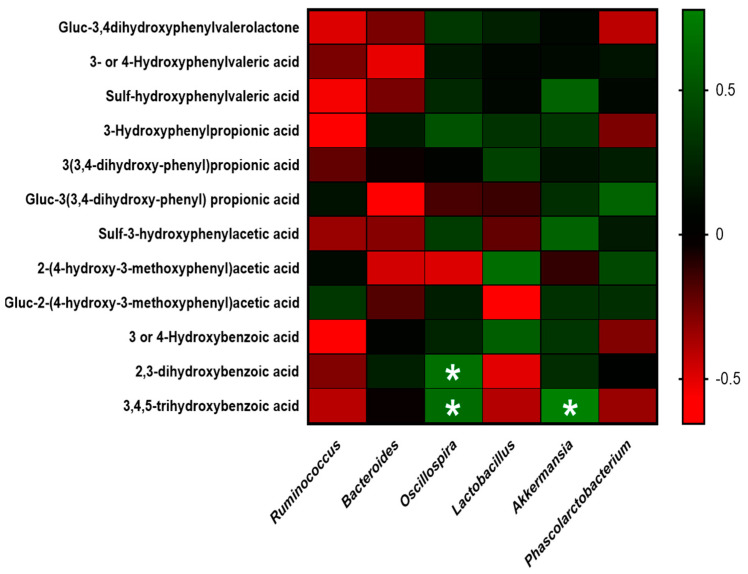
Interactions between gut microbiota genera and polyphenol-derived metabolites. Heatmap of correlation between the main phenolic metabolites in feces of CCB animals and the main significantly altered genera in the gut microbiota. Spearman correlation values were used for the matrix. The intensity of the color represents the degree of association. * Denotes adjusted *p* < 0.05.

**Table 1 antioxidants-12-01519-t001:** Diabetic parameters of Zucker lean rats (ZL), Zucker diabetic rats (ZDF) and Zucker diabetic rats fed with a CCB-rich diet (ZDF-CCB) *.

	ZL	ZDF	ZDF-CCB
Daily food intake (g)	18.55 ± 1.53 ^a^	31.33 ± 2.27 ^b^	29.65 ± 2.13 ^b^
Body weight gain (g)	98 ± 10 ^a^	55 ± 8 ^b^	63 ± 5 ^b^
Glycemia (mg/dL)	78.2 ± 11.3 ^a^	292.0 ± 21.5 ^b^	185.9 ± 25.0 ^c^
Hb1ac (%)	4.6 ± 0.6 ^a^	20.3 ± 4.2 ^b^	16.3 ± 2.0 ^c^
Insulinemia (ng/dL)	0.39 ± 0.06 ^a^	0.37 ± 0.05 ^a^	0.51 ± 0.09 ^b^
HOMA-IR	2.0 ± 0.4 ^a^	9.0 ± 1.8 ^b^	5.4 ± 1.3 ^c^
HOMA-B	178.4 ± 12.7 ^a^	18.4 ± 3.4 ^b^	55.8 ± 10.4 ^c^
AUC (mg/dL/min)	18,180.0 ± 2128.7 ^a^	54,440.6 ± 4082.7 ^b^	31645.0 ± 10,021.1 ^c^

* AUC= Area under the curve calculated from the blood glucose levels during glucose tolerance test. Data represent the means ± SD of 6–8 animals. Means in a row without a common letter differ; *p* < 0.05.

**Table 2 antioxidants-12-01519-t002:** Levels of phenolic metabolites in the feces of Zucker lean rats (ZL), Zucker diabetic rats (ZDF), and Zucker diabetic rats fed with a CCB-rich diet (ZDF-CCB) *.

Class	Compound	ZL nmol/g (n)	ZDF nmol/g (n)	ZDF-CCB nmol/g (n)
Valerolactones	Gluc-3,4dihydroxyphenylvalerolactone	nd (-)	nd (-)	19.0 ± 3.2 ^a^ (5)
Phenylvaleric acid	3- or 4-hydroxyphenylvaleric acid	nd (-)	nd (-)	302 ± 65 ^a^ (5)
	Sulf-3,4hydroxyphenylvaleric acid	nd (-)	nd (-)	82.1 ± 28.7 ^a^ (2)
Phenylpropionic acids	3-hydroxyphenylpropionic acid	nd (-)	nd (-)	117 ± 30 ^a^ (6)
	3(3,4-dihydroxy-phenyl) propionic acid	nd (-)	nd (-)	59.7 ± 6.4 ^a^ (2)
	Gluc-3(3,4-dihydroxy-phenyl) propionic acid	nd (-)	nd (-)	17.8 ± 0.4 ^a^ (3)
Phenylacetic acids	Sulf-3-hydroxyphenylacetic acid	nd (-)	nd (-)	0.9 ± 0.3 ^a^ (3)
	2-(4-hydroxy-3-methoxyphenyl) acetic acid	nd (-)	nd (-)	0.8 ± 0.1 ^a^ (5)
	Gluc-2-(4-hydroxy-3-methoxyphenyl) acetic acid	nd (-)	nd (-)	0.3 ± 0.01 ^a^ (4)
Benzoic acids	3 or 4-hydroxybenzoic acid	nd (-)	nd (-)	4.8 ± 0.4 ^a^ (8)
	2,3-dihydroxybenzoic acid	2.9 ^b^ (1)	4.8 ± 0.3 ^a^ (2)	5.2 ± 0.8 ^a^ (8)
	3,4,5-trihydroxybenzoic acid	nd (-)	nd (-)	9672 ± 2968 ^a^ (6)

* Data are expressed as mean ± SD and means with different letters differ (*p* < 0.05). n means the number of animals in which the metabolite was detected in each group.

## Data Availability

The data presented in this study are available in the article.

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
