# Peer review of "Dietary Supplementation with a Cocoa–Carob Blend Modulates Gut Microbiota and Prevents Intestinal Oxidative Stress and Barrier Dysfunction in Zucker Diabetic Rats"

_antioxidants, 2023, doi:10.3390/antiox12081519_

Round 1

Reviewer 1 Report

In present study, García-Díez E. et al. put efforts to provide a double-edge sword interactions of polyphenols-enroched CCB extract and gut microbiota in animal models of DM2. The article is scientifically sound, though some considerable changes are suggested to improve the presentation of work.

1. Line 92. What it means by (12)?

2. Line 106. Use appropriate unit for temperature.

3. Table 1: ". AUC= Area under the curve. Data represent the means ± SD of 6-8 animals. Means in a row without a common letter differ, P < 0.05".

Write these sentences in the table footnotes. Also see the other tables for same revision.

4. Write "Firmicutes/Bacteroidetes" in Italic format.

5. Hence, the study showed reciprocal interactions between CCB polyphenols and gut microbiota, it is important to mention the polyphenolic profile of the extract, perhaps the detailed polyphenolic profile has been performed in reference 17. It is important to mention it in the Discussion under section 3.4, so it will be easy to understand to the link between polyphenols and their metabolites.

6. Most importantly, it is important to define the novelty in the current study, as the reciprocal interaction of polyphenols and gut microbiota is already a knwon concept and in the current study only the source of polyphenols has been changed i.e., CCB. The same is already reported in number of studies with different dietary sources of polyphenolic compounds.

7. The streangths and limitation of the present study need to be mentioned.

Author Response

We would like to thank the Reviewer for the careful reading of the manuscript, as well as for the encouraging comments and helpful suggestions. All comments and suggestions have been accepted and we believe that the manuscript has been largely improved by the changes introduced.

  1. Line 92. What it means by (12)?

            We would like to apologize for the mistake regarding the reference in the original manuscript. We have rectified this error in the revised manuscript by replacing it with the correct reference (17), which accurately describes the composition of CCB.

  1. Line 106. Use appropriate unit for temperature.

            Following the referee advice, we have use the appropriate unit for temperature.

  1. Table 1: "AUC= Area under the curve. Data represent the means ± SD of 6-8 animals. Means in a row without a common letter differ, P < 0.05". Write these sentences in the table footnotes. Also see the other tables for same revision.

            In agreement with the Referee, these sentences have been written in the table footnotes (tables 1 and 2) of the revised manuscript.

  1. Write "Firmicutes/Bacteroidetes" in Italic format.

            Following the referee advice, we have written “Firmicutes” and “Bacteroidetes in Italic format.

  1. Hence, the study showed reciprocal interactions between CCB polyphenols and gut microbiota, it is important to mention the polyphenolic profile of the extract, perhaps the detailed polyphenolic profile has been performed in reference 17. It is important to mention it in the Discussion under section 3.4, so it will be easy to understand to the link between polyphenols and their metabolites.

            In total agreement with the referee, we have introduced a new paragraph in the discussion (under section 3.4) to indicate the polyphenolic profile of CCB, aiming to understand the link between polyphenols and their metabolism (lines 477-489).

  1. Most importantly, it is important to define the novelty in the current study, as the reciprocal interaction of polyphenols and gut microbiota is already a knwon concept and in the current study only the source of polyphenols has been changed i.e., CCB. The same is already reported in number of studies with different dietary sources of polyphenolic compounds.

            In total agreement with the reviewer, we have included a sentence at the end of the discussion (lines 508-512) in order to highlight the novelty of the present study for the reader.

  1. The strengths and limitation of the present study need to be mentioned.

            According to the reviewer, a paragraph addressing the potential strengths and limitations has been included at the end of the discussion (Lines 512-520).

Reviewer 2 Report

The authors concluded that cocoa-carb blend (CBB) ameliorates intestinal impairment in over-feeding-induced diabetes in Zucker Diabetic Fatty (ZDF) rats. Lines of results shown in the manuscript may catch readers. However, there needs to be more information.

I have some comments on this manuscript below.

1.       Information on CBB is lacking. The authors referred to the paper (12), but there is no information. This implies improper citation. If the authors did not provide information on CBB, we can not understand the effects of CBB on intestinal disorders in diabetic rats.

2.       Daily diet intake, changes in body weight, and changes in the blood glucose in OGTT are essential information.

3.       Pictures and letters of data are too small. Please add arrows or something to locate important points in the pictures and add information.

4.       The authors mentioned the lack of correlation between hemoglobin A1C and bacterial groups. Is it has nothing to relate with long-term glycemic control? Please discuss it.

5.       If the authors want to conclude that the protective effects of CBB were associated with the modulation of the gut microbiota, the authors need to investigate fecal transplantation experiments. It is also unclear whether the effects of polyphenols in feces can potentially improve intestinal disorders in diabetic rats.

6.       The title is challenging to understand. Please develop a title that makes it easy to understand that CBB feeding improves intestinal disorders in diabetic rats.

Author Response

We would like to thank the Reviewer for the careful reading of the manuscript, as well as for the encouraging comments and helpful suggestions. All comments and suggestions have been accepted and we believe that the manuscript has been largely improved by the changes introduced.

  1. Information on CBB is lacking. The authors referred to the paper (12), but there is no information. This implies improper citation. If the authors did not provide information on CBB, we can not understand the effects of CBB on intestinal disorders in diabetic rats.

            We would like to apologize for the mistake regarding the reference in the original manuscript. We have rectified this error in the revised manuscript by replacing it with the correct reference (17), which accurately describes the composition of CCB.

  1. Daily diet intake, changes in body weight, and changes in the blood glucose in OGTT are essential information.

            We fully agree with the Referee's suggestions. In response, we have incorporated the information regarding daily diet intake and changes in body weight in Table 1 of the revised manuscript. Furthermore, we want to clarify that changes in glucose values during the glucose tolerance test (GTT) were already included in Table 1 as the area under the curve (AUC). The AUC was calculated from the blood glucose levels obtained during the GTT. We have now provided this clearer explanation in the text (lines 309-317).

  1. Pictures and letters of data are too small. Please add arrows or something to locate important points in the pictures and add information.

            Following the reviewer advices, we have augmented the size of the letters in the figures. Additionally, we have included arrows in the histological pictures (Figures 1 and 2) to help in the precise localization of important points.   

  1. The authors mentioned the lack of correlation between hemoglobin A1C and bacterial groups. Is it has nothing to relate with long-term glycemic control? Please discuss it.

            This is an interesting point. In fact, while the gut microbiota undoubtedly has an impact on metabolic health, including glycemic control, the relationship between specific bacterial groups and HbA1c levels remains complex and not yet fully elucidated. Some studies have reported significant associations between certain bacterial species or groups and HbA1c levels, but others have found conflicting results or limited correlations. In this sense, it is important to consider the broader context and multiple factors when evaluating long-term glycemic control in individuals with diabetes. Previous studies have highlighted the influence of individual variations on the relationship between gut microbiota and glycemic control (Hermes et al., Sci Rep. 2020, 10, 7523). Furthermore, the different stages of diabetes can also contribute to variations in gut microbiota composition and its impact on the glycemic control (Letchumanan et al., Front Cell Infect Microbiol. 2022, 12, 943427). Additionally, the inherent heterogeneity of the gut microbiota itself adds more complexity to this relationship (He et al., Curr Microbiol, 2023, 80, 132). Considering all these factors, it is clear that further research is needed to fully understand the intricate relationship between gut microbiota and long-term glycemic control.

This concern has been now included in the discussion (lines 449-456) of the revised version of the manuscript.

  1. If the authors want to conclude that the protective effects of CBB were associated with the modulation of the gut microbiota, the authors need to investigate fecal transplantation experiments. It is also unclear whether the effects of polyphenols in feces can potentially improve intestinal disorders in diabetic rats.

            We totally agree with the concern of the referee. We also consider that fecal transplantation experiments could be a very reliable tool to dissect association from causality and we will take into consideration the referee´s suggestion for further studies. Using fecal transplantation from CCB fed donors could provide valuable insights into the role of the gut microbiota in mediating the beneficial effects of CCB. This concern has been now included as a limitation of the study in the discussion section (lines 515-518).

            Regarding the second question, the presence of phenolic-derived metabolites in feces is a strong indicator of the transformation of polyphenols by the gut microbiota. Since these metabolites are not either immediately generated after reaching the colon or excreted after ingested, they can led to different biological activities. First, they may exert important health effects due to their interaction with the colonic tissue and their ability to generate an antioxidant environment. For instance, it was found that moderate wine consumption reduced faecal water cytotoxicity and this was correlated to the presence of microbial-derived phenolic metabolites as the ones identified here, such as phenylpropionic acid (Zorraquín-Peña et al., 2020, Nutrients, 12, 2716).  Additionally, it must be considered that part of these metabolites could have already been absorbed to provoke systemic beneficial effects, as suggested from studies with radiolabelled flavanols where evidences of enterohepatic recirculation were observed (Stoupi et al., 2011, Drug Metabolism Disposition, 38, 287–291). In the same way, significant correlations have been reported between the fecal and urine levels of dihydroferulic acid, a microbial-derived phenolic metabolite, indicating absorption and further recirculation of the compound (Vitaglione et al., 2015, 101, 251-61).

  1. The title is challenging to understand. Please develop a title that makes it easy to understand that CBB feeding improves intestinal disorders in diabetic rats.

            The tittle of the manuscript has been appropriately changed according to the indications of the Reviewer.

Round 2

Reviewer 1 Report

The authors have made efforts to improve the overall wquality of the MS and to address all the review comments.

However, I have aminor concenr, in replies to review comments # 6 and 7, what authors refering to in Lines 508-5112 and 512-520, respectively. It ois suggested to indictae the corrent line numbers so I can access the revision for both of these comments properly.

Author Response

We would like to apologize for the confusion regarding the lines indicated in comments 6 and 7.
In the latest version of the manuscript (revised manuscript) the correct lines are:
- Comment 6: (lines 447-481)
- Comment 7: (lines (481-489)
These lines correspond to the last paragraph of the Results and discussion section.